# Introducing the crystalline phase of dicalcium phosphate monohydrate

Bing-Qiang Lu [1,7✉], Tom Willhammar [2,7], Ben-Ben Sun[3], Niklas Hedin [2], Julian D. Gale [4] &
Denis Gebauer [5,6✉]

Calcium orthophosphates (CaPs) are important in geology, biomineralization, animal metabolism and biomedicine, and constitute a structurally and chemically diverse class of minerals. In the case of dicalcium phosphates, ever since brushite ($CaHPO_4 \cdot 2H_2O$, dicalcium phosphate dihydrate, DCPD) and monetite ($CaHPO_4$, dicalcium phosphate, DCP) were first described in 19th century, the form with intermediary chemical formula $CaHPO_4 \cdot H_2O$ (dicalcium phosphate monohydrate, DCPM) has remained elusive. Here, we report the synthesis and crystal structure determination of DCPM. This form of CaP is found to crystallize from amorphous calcium hydrogen phosphate (ACHP) in water-poor environments. The crystal structure of DCPM is determined to show a layered structure with a monoclinic symmetry. DCPM is metastable in water, but can be stabilized by organics, and has a higher alkalinity than DCP and DCPD. This study serves as an inspiration for the future exploration of DCPM's potential role in biomineralization, or biomedical applications.

[1] State Key Laboratory of High Performance Ceramics and Superfine Microstructure, Shanghai Institute of Ceramics, Chinese Academy of Sciences, Dingxi Road 1295, 200050 Shanghai, China. [2] Department of Materials and Environmental Chemistry, Stockholm University, SE-106 91 Stockholm, Sweden. [3] Department of Orthopedics, Shanghai Jiao Tong University Affiliated Sixth People's Hospita, 600 Yishan Road, 200233 Shanghai, China. [4] Curtin Institute for Computation, School of Molecular and Life Sciences, Curtin University, PO Box U1987, Perth, WA 6845, Australia. [5] Department of Chemistry, Physical Chemistry, University of Konstanz, Universitätsstraße 10, Box 714, D-78457 Konstanz, Germany. [6] Present address: Institute of Inorganic Chemistry, Leibniz University Hannover, Callinstr. 9, D-30167 Hanover, Germany. [7] These authors contributed equally: Bing-Qiang Lu, Tom Willhammar. ✉email: b.q.lu@mail.sic.ac.cn; gebauer@acc.uni-hannover.de

Calcium orthophosphates (CaPs) are common and important in geological and biological systems[1], constituting a family of compounds with varying Ca/P atomic ratios and structural water contents. Important members are hydroxyapatite $(Ca_{10}(PO_4)_6(OH)_2)$[2], octacalcium phosphate $(Ca_8H_2(PO_4)_6\cdot5H_2O)$[3], brushite $(CaHPO_4\cdot2H_2O)$[4], monetite $(CaHPO_4)$[5], and tricalcium phosphate $(Ca_3(PO_4)_2)$[6,7]. Amorphous calcium phosphates (ACPs) have different chemical formulae, and can be the precursors of various crystalline CaPs[8–13], including hydroxyapatite, in vivo[14–20]. However, the role of ACPs in biomineralization has been debated[21–25]. CaPs are involved in hard tissue formation and the metabolism of humans and animals, whereas they can also form pathologically[13,26].

Due to their chemical composition being similar to that of hard tissues, high biocompatibility and bioactivity, synthetic CaPs have been extensively studied for uses in biomedicine, for example, as coatings of metallic implants and synthetic substitutes for bone repair[27]. Since X-ray crystallography was developed in the early 20th century[28,29], the structures of all known CaPs were resolved in the course of that century, while some CaPs were discovered for the first time even earlier[7]. Despite their structural and chemical variability, however, no further crystalline form of CaP has been found lately. Until now, two different forms of crystalline CaPs with a Ca/P atomic ratio of 1:1 have been documented: $CaHPO_4\cdot2H_2O$ (brushite; dicalcium phosphate dihydrate, DCPD) and $CaHPO_4$ (monetite; dicalcium phosphate, DCP), both playing important roles in geology[4,5] and biology[30,31]. In DCPD, structural water molecules link layers composed of $CaHPO_4$ via hydrogen bonds, whereas DCP is anhydrous and the structure is not layered[32,33] (see below). These differences give rise to rather distinct chemical and physical properties for the two forms[34,35]. DCP and DCPD are both useful in biomedical applications, especially for the curing of calcium phosphate cements in clinical orthopedics. However, DCPD seems to provide generally better performance in real uses, as it exists in the majority of setting products[36,37].

Metastable CaPs with structural water are thus promising for advancing biomedical applications. In this work, we introduce another crystalline form of CaP: dicalcium phosphate monohydrate (DCPM, $CaHPO_4\cdot H_2O$). Due to its layered nature and structural water content, advantageous properties for applications within the biomedical field are anticipated. Also, it might play an important role as an intermediate in CaP biomineralization.

## Results

**Preparation and characterization of DCPM.** Amorphous calcium hydrogen phosphate (ACHP), with a Ca/P atomic ratio of 1.0[38], which is identical to that of DCPM, was used as a precursor for the crystallization of DCPM (see Methods section for details). The amorphous character of its precursor was confirmed by X-ray powder diffraction (XRPD), transmission electron microscopy (TEM), and electron diffraction (ED) (Supplementary Fig. 1). In water-poor environments, that is, mixtures of methanol and water, or in humid air, ACHP transforms into DCPM (see Methods section for details). Large crystals of DCPM, sufficient for single-crystal X-ray diffraction (SCXRD, see Fig. 1b and Supplementary Fig. 2 for typical morphologies), could unfortunately not be obtained. However, ED provides a tool to overcome this issue, as the strong interactions between electrons and matter allow diffraction data to be obtained from crystals that are much smaller than in the case of X-rays. Indeed, recently, single-crystal electron diffraction (SCED) methods have proven to be invaluable for the structure determination of sub-micrometer-sized crystals for a wide range of compounds[39–46]. Three-dimensional SCED data were collected for DCPM by continuously rotating a

crystal in the electron beam of a transmission electron microscope while collecting a series of selected-area electron-diffraction patterns, the so-called continuous Rotation Electron Diffraction technique (cRED), Fig. 1. This procedure was aided by sequential defocusing of the intermediate lens of the TEM in order to confirm the position of the crystal in the beam[47]. The reconstructed three-dimensional reciprocal lattice was successfully indexed using a monoclinic unit cell (space group $P2_1/c$, Fig. 1, Supplementary Figs. 3 and 4, and Supplementary Comment Section). Ab-initio structure determination revealed a layered structure with one unique calcium and one phosphate ion in the asymmetric unit constituting DCPM, while the location of the hydrogen atom on the phosphate could not be determined from this data alone. In addition, one water molecule per calcium and phosphate ion was found, consistent with its chemical formula, which was further confirmed by thermogravimetric analysis (TGA) (Supplementary Fig. 5). Although the SCED data contained diffuse scattered lines, indicating layered disorder (see Fig. 1d and Supplementary Fig. 6), least-squares refinement converged with a residual of 0.260 (see Supplementary Table 1 for details). So as to improve the confidence in the structural model of DCPM obtained by SCED further, and to show that it is representative for the bulk of our sample, Rietveld refinement of XRPD data converged with a $R_{wp}$ of 6.67% (Supplementary Fig. 7, Supplementary Table 2). This yielded a monoclinic unit cell for DCPM with $a = 8.0063(4)$ Å, $b = 6.7954(5)$ Å, $c = 7.7904(5)$ Å, $\alpha = \gamma = 90°$, $\beta = 91.548(4)°$. Due to uncertainties in unit cell parameters as determined from ED data, all refinements were performed using the unit cell from XRPD data. The structure of DCPM refined against cRED and XRPD data shows only minor changes with an average difference in atomic positions of 0.16 Å and maximum difference of 0.25 Å. The coordination environment around Ca and P is the same for the structures after the two refinements, see Supplementary Figs. 8 and 9.

In order to further define the structural characteristics of DCPM, first principles quantum mechanical calculations were performed (see Methods section for full details). Hydrogen atoms were inserted into the structure for a variety of different initial positions for both the $HPO_4^{2-}$ ions and water, followed by geometry optimization and annealing via molecular dynamics. A consistent picture emerges in which two chains of hydrogen bonded $HPO_4^{2-}$ anions run parallel to the $b$ axis, with the OH groups pointing in opposite directions in each chain. Attempts to transfer the proton along the chain (i.e., from OH…O to O… HO) led to spontaneous return of the hydrogen to its initial site, indicating a strong preference for the original order. Water coordinates to one calcium ion via oxygen and is simultaneously able to hydrogen bond to two different anions. All hydrogen positions are commensurate with the space group without the need for partial occupancy or disorder (see Supplementary Fig. 8).

The unit cell parameters of DCPM differ from those of DCPD (monoclinic; $a = 5.812$ Å, $b = 15.180$ Å, $c = 6.239$ Å, $\alpha = \gamma = 90°$, $\beta = 116.42°$) and DCP (triclinic; $a = 6.910$ Å, $b = 6.627$ Å, $c = 6.998$ Å, $\alpha = 96.34°$, $\beta = 103.82°$, $\gamma = 88.33°$). Consistently, also the atomic structures of $CaHPO_4\cdot xH_2O$, with $x = 0$, 1, and 2 for DCP, DCPM, and DCPD, respectively, differ significantly (Fig. 2). Although both DCPM and DCPD contain water molecules located between the layers, the atomic arrangements within each layer of the two structures are different. The structural differences of the three CaPs were further confirmed by XRPD patterns and infrared (IR) spectra (Fig. 3). Regarding the vibrational modes of the $HPO_4^{2-}$ ion (Fig. 3b) in the three forms, the $v_6$, and $v_{6'}$ bands share similar wavenumbers ($\sim1126$ cm$^{-1}$ and $\sim1060$ cm$^{-1}$, respectively), while those of $v_2$, $v_3$, and $v_4$ bands differ (DCPM: 1001 cm$^{-1}$ for $v_2$; 901 and 887 cm$^{-1}$ for $v_3$; 551 and 533 cm$^{-1}$ for $v_4$); the split $v_3$ band in the spectrum of DCPM also contrasts

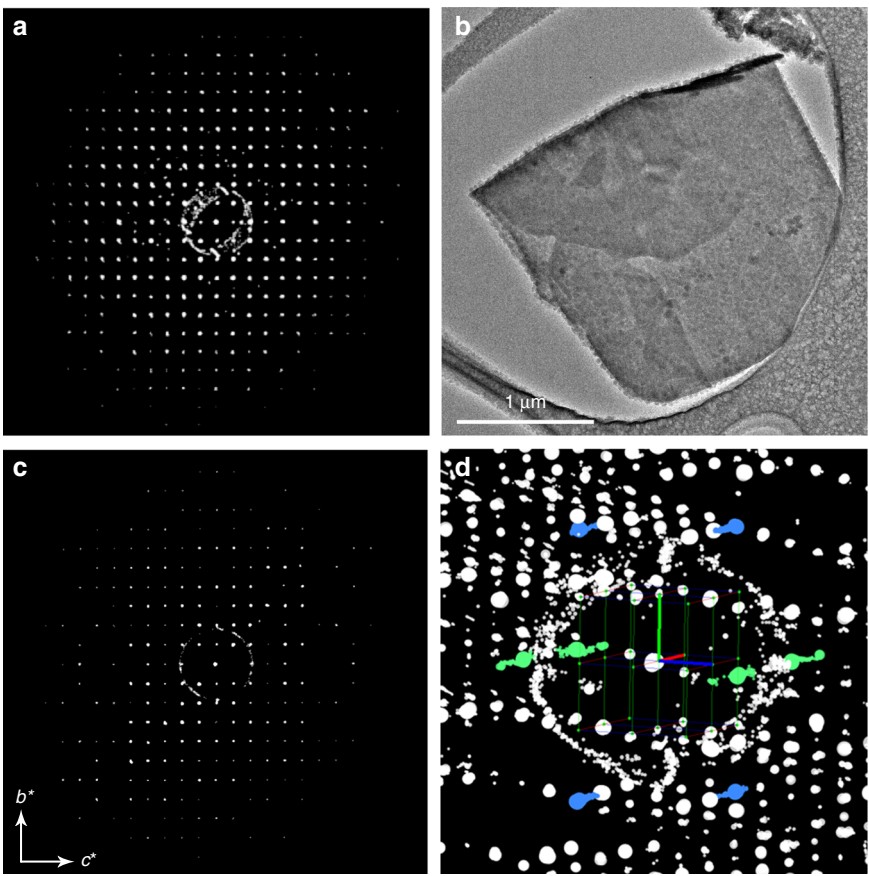

**Fig. 1 Electron microscopy-based structure determination of DCPM. a** The reconstructed three-dimensional reciprocal lattice from continuous rotation electron diffraction data collected from (**b**) a crystal of DCPM can be indexed by a monoclinic unit cell. **c** Section through the reconstructed reciprocal lattice showing the *0kl* family of reflections. Systematic extinctions can be observed for *0k0* and *00l* families of reflections which is consistent with the space group $P2_1/c$. Further details on the symmetry deduction can be found in Supplementary Figs. 3 and 4. **d** Enlarged part of the 3D reconstructed reciprocal lattice shows some diffuse scattering running along the *a\**-axis, here, the *a\**, *b\**, and *c\**-axes are indicated by red, green and blue vectors, respectively. The *121* reflection with surrounding diffuse scattering and its symmetry related equivalents are marked in blue and the *h02* line is marked in green. The diffuse circles in **a**, **c**, and **d** are due to diffraction from small ice crystals formed under liquid nitrogen cooling conditions. Source data are provided as a Source Data file.

DCPD and DCP. Lastly, the $H_1$–$H_4$ bands, which can be assigned to two crystallographically distinct waters in DCPD, are not seen in the spectrum of DCPM. The pattern of the OH stretching modes is further supported by quantum mechanical calculations, which yield two groups of four IR-active modes each for DCPD in the ranges 3477–3429 and 3253–3146 cm$^{-1}$, whereas DCPM only has a single group of four active modes between 3402 and 3364 cm$^{-1}$ consistent with the experimental spectra (Fig. 3b). A structural comparison of DCPM with other CaPs is summarized in Supplementary Table 3.

DCPM was previously neither observed during the dehydration of DCPD[49,50] nor the hydration of DCP in H$_2$O[51,52]. We also failed to obtain DCPM from other amorphous calcium phosphates (with Ca/P atomic ratios higher than 1.0). This indicates that ACHP is an essential precursor for the crystallization of DCPM. Although prepared by ACHP transforming in mixtures of methanol and water, DCPM is an intermediate phase between ACHP and DCP. As shown in Supplementary Fig. 10, DCPM can be formed as early as 2 min, and kept for 7 h. But after 24 h, while minor fractions of DCPM can still be detected by IR spectroscopy and XRPD, the majority phase became DCP.

Crystals of DCPM also formed via crystallization of ACHP in humid air (humidity 95%, 25 °C) during the transformation of ACHP to DCPD (as opposed to DCP that formed in mixed solvent as discussed above). Under these conditions, DCPM was the majority phase of the transformation of ACHP after 4 h (whereas two reflections were indicative of a DCPD impurity). ED of the as-obtained samples confirmed that the structure was identical to those prepared by the mixed water/methanol approach (Supplementary Fig. 11). This identity was evident by comparing the corresponding XRPD patterns and IR spectra of the DCPM from the distinct preparations (Supplementary Figs. 12 and 13). However, DCPM that crystallized from methanol/water mixtures exhibited a higher crystallinity as the broad feature of ACHP in the XRPD pattern was significantly reduced (Supplementary Fig. 12). After treating ACHP for 12 h in humid air, DCPD formed (alongside impurities of OCP, Fig. 4a). That the crystallization of ACHP in humid air progressed via DCPM to DCPD after 4 h and 12 h was corroborated by IR spectroscopy (Fig. 4b). TEM analysis (Fig. 4c) revealed corresponding morphological changes, where the spherical aggregates of the precursor ACHP yielded platelet-like DCPM without obvious elongations after 4 h, alongside some un-crystallized ACHP nanoparticles. After 12 h (Fig. 4c), typical elongated platelets of DCPD with well-defined edges had formed. H$_2$O is required for the crystallization of DCPM from ACHP, as the amorphous precursor did indeed not crystallize in dry air (humidity < 5%, Methods section) for at least 1 month. The presence of too much water will lead to crystallization yielding other phases such as DCPD[38].

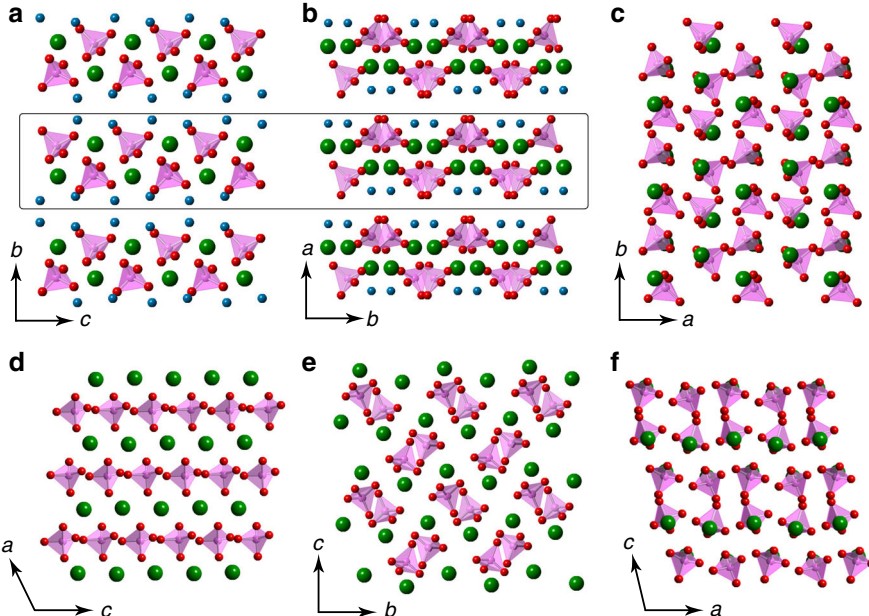

**Fig. 2 Structural comparison of DCPD, DCPM and DCP.** The crystal structures of DCPD (**a**), DCPM (**b**), and DCP (**c**) viewed along *a*, *c*, and *c*-axes, respectively. The layers in DCPD and DCPM are marked in (**a**) and (**b**) and are distinctly different in their structures, as seen in views perpendicular to each of the layers presented in **d** and **e**, respectively. **f** Shows the structure of DCP viewed along the *b*-axis. In the figures, magenta represents P; green: Ca; red: O of the $HPO_4$ ion; blue: O of the water molecules (H atoms are not shown).

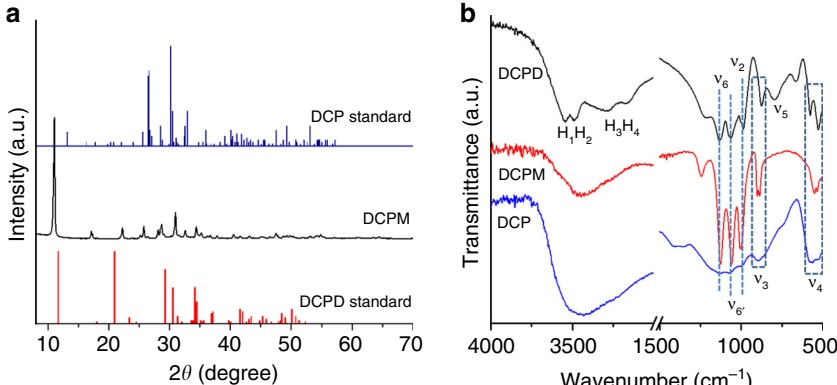

**Fig. 3 Characterization of DCPD, DCPM, and DCP with XRPD and IR spectroscopy. a** Comparison of the XRPD pattern of DCPM with the standard diffractograms of DCPD (JCPDS 09-0077) and DCP (JCPDS 09-0080). **b** The IR spectra of DCPD, DCPM, and DCP. The assignment of vibration modes follows Bailey and Holt[48]. Source data are provided as a Source Data file.

**Properties of DCPM.** In water, DCPM was stable for at least 0.5 h, but transformed into poorly crystallized HAP within 1 h (Fig. 5a), which is the main biomineral component of human bones and teeth. In contrast, DCP and DCPD did not transform within 2 h (Supplementary Fig. 14), although they will ultimately yield more stable CaPs[52]. Organics, such as citrate salts, which are abundant in vivo and biocompatible[53] can effectively stabilize DCPM for at least 2 h (Fig. 5a); polyelectrolytes such as sodium polyacrylate (NaPAA) can stabilize it for even longer times, i.e., at least 13 h (Fig. 5a). Notably, upon initial dissolution, the pH of the DCPM solution can reach a value of 8.2 before DCPM transforms (Fig. 5b), which is the highest of the three forms (DCP, DCPM, and DCPD). The comparably high pH in relation to the aqueous chemistry of the different protonation states of the phosphate anions upon $HPO_4^{2-}$ dissolution indicated that DCPM has the highest solubility among the three CaPs. Support for this comes from theoretical calculations (see Supplementary Table 4), which indicate that formation of DCPM from monetite

is as endergonic as formation of brushite, though the latter has twice as much water. A brief comparison of DCPM with other CaPs in terms of chemical properties is compiled in Supplementary Table 5. It is also interesting that DCPM exhibited a significantly higher adsorption capacity for some organic molecules compared to DCP and DCPD, e.g., methyl blue, congo red, doxorubicin hydrochloride (anti-cancer drug), and ibuprofen (analgesic drug) (Supplementary Table S6). This higher adsorption capacity, in combination with the biocompatibility shown in Supplementary Fig. 15, points toward a significant potential use of DCPM for improving drug delivery carriers or pollution treatment materials. In their totality, in any case, these properties highlight that DCPM is a material that may significantly improve or extend current in vivo applications of CaPs.

## Discussion

In conclusion, the CaP family has been expanded by DCPM, exhibiting a unique crystallographic structure. Considering the

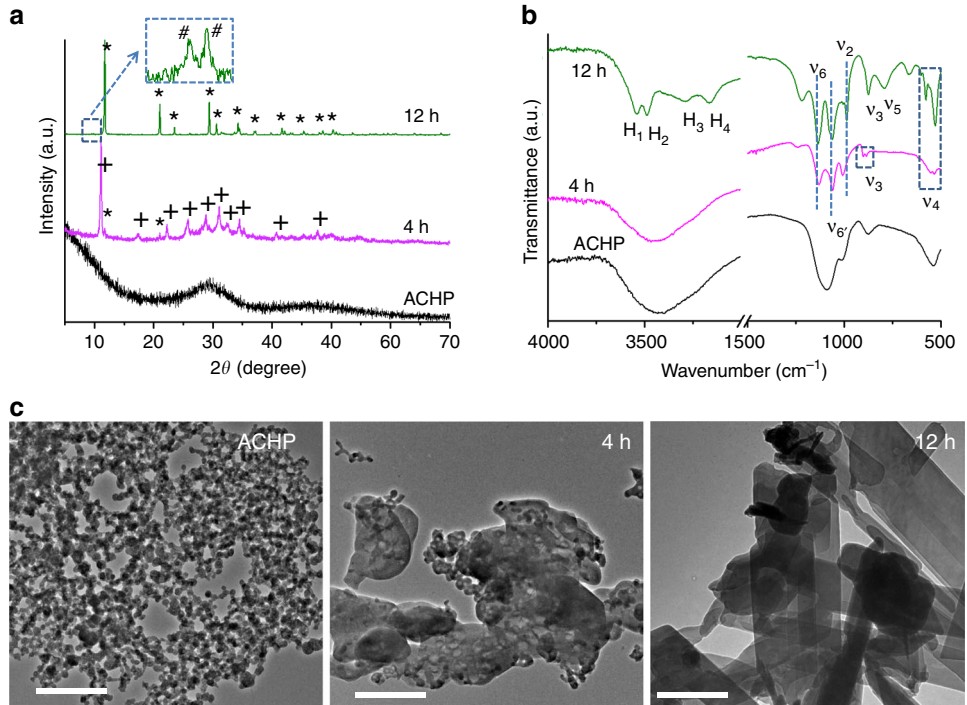

**Fig. 4 Formation of DCPM from ACHP in wet air.** XRPD patterns (**a**), IR spectra (**b**) and TEM micrographs (**c**) of ACHP in wet air (25 °C, 95% humidity) after different times as indicated. In the XRPD patterns (**a**), the symbols *, + and # mark the diffraction peaks of DCPD, DCPM; and OCP, respectively. The inset magnifies the diffractogram of the sample (12 h) in the range of 8–10° 2θ. In the IR spectra (**b**), "$\nu_i$" represent different vibrational modes i of the phosphate ion; $H_1$–$H_4$ indicate the $\nu_{(OH)}$ modes of four different hydrogen bonds of structural $H_2O$. The assignment of vibration modes follows Bailey et al.[48]. All scale bars: 500 nm. Source data are provided as a Source Data file.

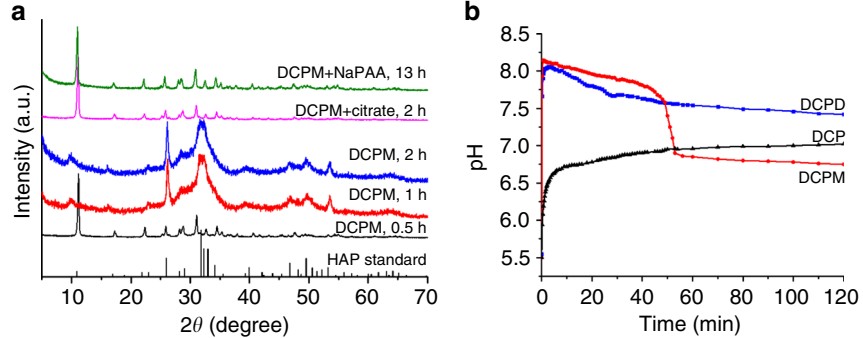

**Fig. 5 Transformation of DCPM in water.** XRPD patterns of particles (**a**) and corresponding pH change (**b**) retrieved after dispersing DCPM in water for different times. The sample obtained after 0.5 h shows the typical XRPD pattern of DCPM, indicating it did not transform during this time. Those obtained after 1 and 2 h show poorly crystallized HAP when compared to the HAP standard diffractogram shown at the bottom (JCPDS 09-0432). The pink and green curves in **a** were obtained from samples incubated in sodium citrate solution (pink) or NaPAA (green) for 2 h and 13 h, respectively, revealing the typical pattern of DCPM, indicating that citrate and NaPAA can significantly stabilize DCPM against transformation. Source data are provided as a Source Data file.

important biological role of CaPs and their wide use in biomedicine, it seems highly promising to study and explore the potential use of DCPM in biomedical applications. Last, but not least, it should be carefully investigated to confirm whether or not DCPM occurs (transiently) in geological or biological natural contexts, e.g., biomineralization.

## Methods
DCPM was obtained either via transformation of ACHP in mixed solvents of methanol and deionized water or in humid air.

**Chemicals**. $Na_3PO_4·12H_2O$ and $Na_2HPO_4$ were purchased from Sigma-Aldrich; methanol, ethanol, DCPD and $CaCl_2$ were purchased from Sinopharm Chemical Reagent Co.; sodium polyacrylate (NaPAA, 50% solution, $M_w$ 3000–5000) was purchased from Aladdin Reagent. All chemicals were used as received without further purification. All of the chemicals were of analytical grade. DCP was prepared by heating DCPD at 195 °C for 1 h and was confirmed by XRPD. Deionized water was used in all experiments.

**Preparation of ACHP**. At ambient temperature (ca. 25 °C), $Na_2HPO_4$ (0.95 mmol) and $Na_3PO_4·12H_2O$ (0.05 mmol) was dissolved in 3 mL deionized water, then quickly added into 2 mL $CaCl_2$ (1.67 mmol) aqueous solution under vigorous stirring. Immediately, the white product of ACHP was formed in the solvent. After 3 s, 80 mL methanol was added to quench the reaction. Then, ACHP was separated from the solvent by centrifugation, washed with methanol 3 times, and dried in a vacuum desiccator. The ACHP obtained has a Ca/P atomic ratio of 1.02, formula $CaHPO_4·H_2O$, confirmed by inductively-coupled plasma optical emmission spectroscopy (ICP-OES) and TGA. Its amorphous character was confirmed by XRPD and ED (Supplementary Fig. 1).

**Preparation of DCPM in a water-methanol mixed solvent**. Typically, the ACHP obtained as described above was quenched in 32.5 mL methanol (not 80 mL), and kept in the solvent (that is, 32.5 mL methanol + 5.0 mL water) which was placed in an oscillating incubator (XZQ-X160, Suzhou Peiying) at constant temperature (90 rpm, 25 °C) for 20 min. Afterwards, the resulting product was centrifuged and washed with ethanol 3 times, and dried in a vacuum desiccator to obtain DCPM. To prevent its transformation, DCPM can be stored in dry (humidity < 5%) or cold (<−20 °C) environments.

**Preparation of DCPM in humid air**. Following the procedure for ACHP preparation, the dried ACHP was incubated for a specified period of time (as indicated below) at 25 °C and 95% humidity with air blowing in a temperature humidity chamber (HWS-080, Shanghai Jinghong). During the first 4 h, the ACHP was stirred with a glass stick for 10 s after every 20 min to ensure that the ACHP contacted the humid air more uniformly.

**The stability of ACHP in dried air**. Five gram $CaCl_2$ powder was sealed in a beaker to create a dried air atmosphere (humidity < 5%), alongside a powder of ACHP (0.050 g) that was kept for 1 month. After the ACHP was again taken out, XRPD patterns showed no apparent reflections indicative of crystallization.

**Transformation of DCPM in water**. 10.0 mg freshly prepared DCPM was placed in a centrifuge tube containing 20 mL deionized water (25 °C). Afterwards, it was kept in an oscillating incubator (XZQ-X160, Suzhou Peiying) for specified times at constant temperature (150 rpm, 25 °C). Then the samples were isolated by centrifugation, washed with ethanol 3 times, and dried in a vacuum desiccator.

In order to study the ability of citrate salts stabilizing DCPM, 15.0 mg trisodium citrate dihydrate was dissolved in water before adding DCPM, while the other conditions remained unchanged.

In order to study the ability of NaPAA towards stabilizing DCPM, 23 mg NaPAA was dissolved in water before adding DCPM (pH adjusted to 7.0 using 1:10 hydrochloric acid), while the other conditions described above remained unchanged.

So as to investigate the pH changes during DCPM transformation in water, 10.0 mg freshly prepared DCPM was dispersed in 20 mL deionized water. Under vigorous stirring and maintaining the temperature at 25 °C, a pH meter (A 600, Asone) was used for real-time monitoring of the pH development in the solution.

DCP and DCPD were used as the controls with the same experimental conditions as DCPM.

**Adsorption capacity of DCPM for organics**. Methyl blue, congo red, doxorubicin hydrochloride (anti-cancer drug) and ibuprofen (analgesic drug) were used to study the adsorption capacities of DCPM. Briefly, at 25 °C, 10.0 mg DCPM was dispersed in water, methanol or ethanol with 2.0 mg mL$^{-1}$ organics (doxorubicin 1.0 mg mL$^{-1}$), and then placed in an oscillating incubator at constant temperature (150 rpm, 25 °C). The incubation was performed for 30 min (aqueous solution) or 2 h (methanol or ethanol solution) so as to adsorb organics on DCPM without phase transformation. Afterwards, the samples were isolated by centrifugation. The concentration of the organic molecules in the supernatant was measured with a UV-Vis spectrophotometer (UV2300II, Techcomp). As the detection of ibuprofen in ethanol solution is problematic, 1 mL of the supernatant was dried in vacuum, then re-dissolved in the same volume of hexane for the UV-vis measurement. The wavelengths for the determination of the concentrations of methyl blue in water, methyl blue in methanol, congo red, doxorubicin hydrochloride and ibuprofen were 596 nm, 591 nm, 501 nm, 495 nm, and 220 nm, respectively. DCP and DCPD were used as the controls with the same experimental procedure.

**Cell viability**. Sprague-Dawley (SD) rat bone marrow mesenchymal stem cells (rBMSCs) were kindly provided by Stem Cell Bank, Chinese Academy of Sciences, catalog number: SCSP-402. Samples of DCP, DCPM and DCPD dispersed in ethanol were added into 24-well plates. After drying and sterilizing by UV, rBMSCs in 1 mL culture medium was added for co-incubation ($0.5 \times 10^5$ cells/well) at 37 °C. After a certain time, the culture medium was removed, and 1 mL fresh medium containing 10% Cell Counting Kit-8 (CCK-8; Dojindo) solution was added to each well. After further incubating for 2 h, 100 μL solution from each well was transferred to a 96-well plate. The absorbance of the sample was measured with a microplate reader (BioRad 680, USA) at 450 nm to determine the cell viability. The cell incubated without samples was used as the control.

**Electron microscopy**. TEM micrographs were obtained on a Zeiss Libra 120 operated at 120 kV. Samples were dispersed in ethanol, then a drop of the dispersion was added onto a carbon-coated Cu grid. The TEM analyses were performed immediately after drying.

**X-ray powder diffraction**. The XRPD patterns were recorded using an X-ray diffractometer (Bruker D8 ADVANCE) in reflectance Bragg-Brentano geometry employing Ni filtered Cu Kα line focused radiation ($\lambda = 1.54178$ Å) at 3000 W (40 kV, 40 mA) power. Each sample was measured with a $2\theta$ rate of 10°/min for

normal characterization, and 1°/min for refinement. The Rietveld refinement was performed using the software Topas version 5. The refinement was performed using isotropic atomic displacement parameters, a pseudo-Voigt peak shape function, a fourth order spherical harmonic correction to take into account preferred orientation, and seven parameters to fit the background. No restraints on the geometry of the structure were used. Details of the structure after Rietveld refinement are available from the Cambridge Crystallographic Data Center with deposition code CCDC 1961467.

**Infrared spectroscopy**. Fourier transform IR spectroscopy was performed on a PerkinElmer spectrum 100 Spectrophotometer. Before measurements, thin KBr disks with a diameter of 1.0 cm were prepared by pressing KBr powder by applying a pressure of 10 MPa. The samples were dispersed in ethanol, then dropped on the KBr disks. After the ethanol was dried, the samples were adsorbed on individual KBr disks. The disks were then used for the measurements. In order to achieve a sufficient signal-to-noise ratio, 16 scans were accumulated for each spectrum.

**Thermogravimetric analyses**. TGA measurements were performed using a thermal analyzer instrument (NETZSCH STA 449 F3 Jupiter®) in a nitrogen atmosphere. The heating rate was 10 °C min$^{-1}$. 10–15 mg of powdered samples were placed in a crucible for measurements, whereas an empty one served as reference.

**ICP-OES**. The elemental composition of the ACPs was determined using an ICP-OES (Horiba, JY2000-2). The samples were dissolved in 1% $HNO_3$ solution for measurements. 5 solutions, containing a different concentration of Ca and P in each one, were prepared for the setup calibration.

**Continuous rotation electron diffraction**. The sample was crushed in an agate mortar, dispersed in absolute ethanol and ultrasonicated for 5 s. Then, a droplet of the suspension was transferred onto a copper grid covered by a holey carbon film. The grid was mounted to a single-tilt cryo-transfer tomography holder, Gatan Type 914, operated at liquid nitrogen cooling at ~−176 °C. The data were collected using a JEOL JEM-2100 TEM operated at an accelerating voltage of 200 kV. The cRED data were collected by continuously tilting the goniometer with a tilt speed of 1.12°/s. During tilt the crystal was tracked by sequential defocusing of the intermediate lens using the software Instamatic[47]. The diffraction patterns were collected using the high-speed hybrid detection camera Timepix Quad (ASI). The datasets were processed using X-ray Detector Software (XDS) in order to extract intensities for structure solution and refinement. The reconstructed reciprocal lattice could be indexed by a monoclinic unit cell and systematic extinctions indicated that the space group was $P2_1/c$, see Supplementary Fig. 4. The structure of DCPM was solved using the software SHELXT[54] and the subsequent least-squares refinement was performed in SHELXL-97[55] using atomic scattering factors for electrons extracted from SIR2014[56]. The structure solution directly identified 7 peaks identified as 1 Ca, 1 P, and 5 O. The least squares refinement converged without the use of any restraints and with reasonable isotropic atomic displacement parameters to an R1 residual of 0.260, using 29 refinement parameters. No restraints on the geometry of the structure were used during the refinement. Details regarding the cRED data collection, as well as the refinement, can be found in Supplementary Table S1. Details of the structure after refinement against cRED data are available from the Cambridge Crystallographic Data Center with deposition code CCDC 1961466. cRED data were also collected from the sample of DCPM prepared in humid air. Structure solution resulted in the same structure as for the sample prepared by mixed solvent and the refinement converged with an R1 residual of 0.256.

**First principles calculations**. All quantum mechanical calculations have been performed within the framework of Kohn-Sham density functional theory using one of two different methods:

1. Geometry optimizations and phonon calculations were conducted using the planewave pseudopotential method as implemented within the code CASTEP version 18.1[57]. On-the-fly generated ultrasoft pseudopotentials were used with a small core for Ca and a planewave cutoff of 1000 eV for a high level of convergence ($dE_{tot}/d\log(E_{cut}) < 0.001$ eV per atom). Reciprocal space was sampled with a Monkhorst-Pack mesh with a spacing of 0.02 Å$^{-1}$. The PBE exchange-correlation functional[58] was used with Grimme's D2 dispersion corrections[59]. All geometries were optimized with a fully variable unit cell to an energy tolerance of $10^{-5}$ eV, a force tolerance of 0.02 eV/Å and a stress tolerance of 0.1 GPa. Phonon calculations were performed at the gamma point only using finite differences, from which the zero point energy and thermal corrections were estimated at 298.15 K. All phases were confirmed to have no imaginary modes at the zone center. Calculations on monetite, brushite and DCPM all used unit cells containing four formula units and so any systematic error due to the use of only the gamma point phonons will be largely canceled. In order to estimate the thermodynamics of water incorporation into the phases, calculations were also performed for a proton-ordered structure of ice-Ih at the same level of theory. Here the

computed hypothetical free energy of ice at standard conditions was corrected using the experimental free energy of fusion under the same conditions to estimate the corresponding thermodynamics of liquid water.

2. Ab initio molecular dynamics was also performed for DCPM in order to explore the stability of the initial hydrogen position choices after geometry optimization. Here the Gaussian-augmented planewave method was used as implemented in the Quickstep[60] module of CP2K[61]. In order to also verify that the computed behavior of DCPM was not especially sensitive to the choice of functional, these calculations were performed using BLYP[62,63] with D3 dispersion corrections[64]. The Goedecker-Teter-Hutter pseudo-potentials[65] and basis sets of TZVP quality were used, except for Ca which used a DZVP basis. A planewave basis with a cutoff of 300 Ry was used for the electron density. Molecular dynamics was performed using a $2 \times 2 \times 2$ supercell of DCPM containing 362 atoms in the NPT ensemble, again with a fully flexible unit cell. A slightly elevated temperature of 330 K was used to enhance the rate of exploration of configuration space, with the use of a canonical sampling velocity rescaling thermostat and barostat[66], both with time constants of 50 fs, and a time step of 0.5 fs. Hydrogen was also assigned a mass of 2 amu (i.e., deuterated). A total of 34 ps of MD was run during which no proton transfer events or significant changes in the orientation of water molecules beyond libration were observed. Quenching of structures harvested from the dynamics resulted in the same structure as obtained initially by geometry optimization.

**Force field calculations**. As a further test of the validity of the DCPM structure, calculations were also performed using the GULP code[67] and a recently fitted force field that was designed to reproduce the properties of aqueous calcium phosphate systems during crystallization[68]. Again the DCPM structure was optimized and a full phonon analysis performed across the Brillouin zone that revealed no instability within the initial space group.

**Reporting summary**. Further information on research design is available in the Nature Research Reporting Summary linked to this article.

## Data availability
Rietveld refinement and cRED data are available from the Cambridge Crystallographic Data Center with deposition code CCDC 1961467 and CCDC 1961466, respectively. The source data underlying Figs. 1b, 3a-b, 4a-c, 4c, 5a-b and Supplementary Figs. 1, 2, 4, 5a, b, 7, 10a, b, 12, 13, 14, 15, and Supplementary Table 6 are provided as a Source Data file. Data is also available from the corresponding author upon reasonable request.

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

## Acknowledgements

B.Q.L. acknowledges financial support by the CSC-DAAD postdoc program for a postdoctoral stay at the University of Konstanz, and characterization help from Sheng-tong Sun and Min Ju. D.G. was a Research Fellow of the Zukunftskolleg of the University of Konstanz, and partly supported by the DFG-funded SFB 1214 of the University of Konstanz during this work. T.W. acknowledges a grant from the Swedish research council (VR, 2014-06948). J.D.G. acknowledges the support of the Australian Research Council through grant FL180100087, as well as the provision of computing resources by the Pawsey Supercomputing Centre and National Computational Infrastructure. We also acknowledge financial support from the Knut and Alice Wallenberg Foundation through the project grant 3DEM-NATUR (no. 2012.0112).

## Author contributions

All authors contributed to data analysis, and paper writing; B.Q.L. designed the study, synthesized the samples and performed the characterizations; T.W. conducted the TEM and ED measurements and determined the crystal structure of DCPM; B.B.S. conducted the cell experiments; N.H. performed additional characterizations; J.D.G. performed the theoretical calculations; D.G. designed and supervised the study, organized experiments, data analysis, paper writing and revisions.

## Competing interests

The authors declare no competing interests.
