## [Peer Review File · Nature Communications]

Reviewers' Comments:

Reviewer #1:

Remarks to the Author:

The paper "Introducing Konhaitite: Dicalcium Phosphate Monohydrate" provides the synthesis and crystal structure analysis of a new phase of a hydrated calcium phosphate. As the authors state, the calcium orthophosphate system includes several phases, as hydrates and anhydrates. The existence of a phase flanked by brushite and monetite is not astonishing. More interesting is the transformation into hydroxyapatite in water. The claim to use the new phase in biomedical applications is quite speculative and unfortunately not supported yet experimentally. Taking the following suggestions into account, especially clarifying the structural features not discussed so far, the paper will be a valuable contribution to the class of orthophosphates. I recommend to submit this work to another journal like Inorganic Chemistry.

Suggestions:

The name 'konhaitite' suggests a mineral name and its approval by the IMA. The suffix '-ite' is reserved for minerals and rocks found in nature. Hence, the authors should remove this label and use the proper chemical name only.

The chosen synthesis route using ACP as precursor provides crystalline DCPM. The crystal structure, successfully solved by continuous rotation electron diffraction, with an R value of 26% shows a layered structure as well indicated by the plate-like shape of the crystals. The electron diffraction data shown in Figure 1 indicates that the structure presented in this paper is not described to its full extend.

- Is the space group P21/c correct? In the picture Fig1c the reflection (010) has a relatively high intensity but should be extincted! While this may happen due to multiple scattering Fig. 1d shows the series of reflections h03 and h01, but these should be extincted in P21/c ($h0l: l = 2n$).

Additional questions:

- How many parameters have been refined using 563 independent reflections?
- Uncertainties for a, b, c, beta are missing in Table S1.

Unfortunately, the treatment of the x-ray powder data has as well some flaws.

- The presented X-ray powder diffraction data is typical for a layered structure. Rietveld refinement provided in the SI confirms the structure but the achieved fit is not explaining all intensities leading to an somewhat acceptable residual of 8.6 %. There are smaller peaks in Fig. 5 at e.g. 16.0 °; 18.2 °, 23.7 ° etc, so an impurity that is not being discussed. The areas like 28 to 31 ° do not fit very well. It would be important to use the stacking error model included in Topas to improve FitQuality. Stacking faults as indicated by ED data should definitely be taken into account.

Additional questions:

- Is there a sign for preferential orientation due to the plate-like shape of the crystals? The geometry of the experiment (capillary or flat sample) is not mentioned?
- Uncertainties for a, b, c, beta in the Rietveld Fit are missing in Table S2.
- In supporting info the wavelength 1.54178 Å is stated while in Table S2 it is 1.5406 Å?

Comparing both methods:

- The atomic positions derived by ED refinement and Rietveld refinement are both missing. How similar are the resulting structures? Are there significant differences in the coordination polyhedra? What are the uncertainties of the atomic positions of the APDs in the Rietveld / ED refinement?
- Why is b smaller in the Rietveld refinement than in the ED, while for a and c it is larger (Table S1 and S2)? If it were a pure camera length effect I would expect a consistent trend.

Reviewer #2:

Remarks to the Author:

The authors carried out a huge data analysis. It follows from the conclusion made in the peer-reviewed work that the authors determine the existence of a phase precursor during the mineralization of dicalcium phosphate dihydrate (DCPD) from amorphous calcium phosphate (ACP). In this work, a new form of calcium phosphate (CaP) is introduced, named Konhaitite (after the two places of its discovery, Konstanz and Shanghai), which is dicalcium phosphate monohydrate (DCPM, $\text{CaHPO}_4 \cdot \text{H}_2\text{O}$). This, of course, is important and necessary knowledge for a deeper understanding of the process of biomineralization. Such research deepens our understanding of the processes and mechanisms of calcium phosphates formation and is consistent with current scientific trends. The work was carried out at a high methodological and scientific level, using a complex of modern methods and instruments.

The disadvantages of the article include the following points.

The abstract does not give a complete picture of the study and the content of the article. The authors have studied a large amount of material, but historical references should be included in the introduction. It is more important in the annotation to describe the features of the method for obtaining DCPM material and its main differences from previously applied in the annotation, it is important to show what methods were used to study the resulting structure and its distinctive characteristics. Twice talking about the importance of material at the beginning and end of an abstract is also unnecessary.

The significance of the study conducted by the authors is undeniable, but in the main part, the methods for obtaining and fixing the structure in question are not sufficiently discussed. The investigated phase is a transitional, difficult to fix, therefore, the features of obtaining and research would deepen our understanding of the results. The authors did not specify either the conditions for the creation of this phase, except that it was obtained by the methanol / water approach, nor the fixation conditions to prevent further transitions and the possibility of research. The reasons for choosing this method using a mixture of methanol and water are not substantiated or described the authors also noted that this phase of the DCPM is formed under conditions of high humidity during the transition of ACP to DCPD, but with less crystallinity. It is difficult to draw such conclusions on the basis of Fig. 3, since a very small scale was chosen and the magnifications were not made for all options (only for 12 h.). The unconditional merit of the authors lies in the fact that they use a large number of existing methods and modern instrumentation. However, perhaps, the authors chose an incorrect presentation of the results. The graphs presented in the work are very difficult to compare with the conclusions made by the authors. For logical conclusions based on the data provided, it would be convenient to show the basic structural and chemical characteristics of the newly formed phase in the form of a table in comparison with the previously known DCPD, ACP, and octacalcium phosphate.

Point-by-point replies

Our replies appear indented and italicized, corresponding changes in the manuscript and the SI are highlighted in yellow.

Reviewer #1 (Remarks to the Author):

The paper “Introducing Konhaitite: Dicalcium Phosphate Monohydrate” provides the synthesis and crystal structure analysis of a new phase of a hydrated calcium phosphate. As the authors state, the calcium orthophosphate system includes several phases, as hydrates and anhydrates. The existence of a phase flanked by brushite and monetite is not astonishing. More interesting is the transformation into hydroxyapatite in water. The claim to use the new phase in biomedical applications is quite speculative and unfortunately not supported yet experimentally. Taking the following suggestions into account, especially clarifying the structural features not discussed so far, the paper will be a valuable contribution to the class of orthophosphates. I recommend to submit this work to another journal like Inorganic Chemistry.

We thank the reviewer for the generally positive assessment; however, we beg to strongly differ regarding their opinion on the importance of our findings. There are certainly several crystalline CaPs, and maybe more will be discovered in the future, but it is anything but straightforward to find them and prepare them in sufficiently pure form.

As we pointed out in the introduction, most of the CaPs were first described before the 19th century, and their crystalline structures were determined in the 20th century after the development of X-ray diffraction techniques (Mater. Sci. Eng. C 33, 3085-3110, 2013). Since then, no new crystalline form of CaP has been reported, to the best of our knowledge.

Considering the chemical compositions of brushite ($\text{CaHPO}_4 \cdot 2\text{H}_2\text{O}$) and monetite (CaHPO_4), it is in our opinion rather surprising that DCPM ($\text{CaHPO}_4 \cdot \text{H}_2\text{O}$) has not yet been reported, or even speculated to exist in the literature.

In this manuscript, we present a simple and low-cost method to prepare DCPM —just by mixing salt solutions and storing in solvents for 20 min. Furthermore, we believe it is important to underscore that each of the CaPs mentioned in the manuscript has, for us, as humans, very important applications. For example, HAP is used as coating on implants; brushite ($\text{CaHPO}_4 \cdot 2\text{H}_2\text{O}$) and monetite are used in biomedical cements; TCP is a component of bioceramics. Also, different forms of CaP are very important in biomineralization, pathological phenomena and diseases (e.g., HAP, brushite, OCP). Thus, finding a new crystalline CaP phase, DCPM, which is connected to these minerals via complex transformation pathways that remain to be explored in the future, is in our opinion a finding of utmost importance:

It is clear that this new form, DCPM, will have different chemical and physical properties from the previously known ones, as it has a unique structure, and it is anything but far-fetched to assume that some of them might be advantageous in applications. Indeed, according to data presented in the manuscript, DCPM is a special crystalline form of CaP, potentially exhibiting the highest metastability (despite possessing fewer crystal waters than DCPD!), high alkalinity and special transformation behavior in water.

Regarding the potential application of DCPM in biomedicine, we included the investigation of related basic properties: the transformation behavior in water, pH, biocompatibility and adsorption behavior on organics, which do support the high potential for use. We would like to emphasize that our paper is focused on the discovery,

the formation pathways, and structure determination of DCPM, similar as for the previous case of calcium carbonate hemihydrate (Z. Y. Zou, W. J. E. M. Habraken, G. Matveeva, A. C. S. Jensen, L. Bertinetti, M. A. Hood, C. Y. Sun, P. U. P. A. Gilbert, I. Polishchuk, B. Pokroy, J. Mahamid, Y. Politi, S. Weiner, P. Werner, S. Bette, R. Dinnebier, U. Kolb, E. Zolotoyabko, P. Fratzl. Science 363, 396-400, 2019). Further investigations have to be made in the future, and we, as well as other researchers, are certainly going to do this.

Suggestions:

The name 'konhaitite' suggests a mineral name and its approval by the IMA. The suffix '-ite' is reserved for minerals and rocks found in nature. Hence, the authors should remove this label and use the proper chemical name only. The chosen synthesis route using ACHP as precursor provides crystalline DCPM.

We thank the reviewer for raising this concern. Following the suggestion, we removed the name of Konhaitite, and only left the chemical name dicalcium phosphate monohydrate (DCPM) in the revised manuscript.

The crystal structure, successfully solved by continuous rotation electron diffraction, with an R value of 26% shows a layered structure as well indicated by the plate-like shape of the crystals. The electron diffraction data shown in Figure 1 indicates that the structure presented in this paper is not described to its full extend.

- Is the space group $P2_1/c$ correct? In the picture Fig1c the reflection (010) has a relatively high intensity but should be extincted! While this may happen due to multiple scattering Fig. 1d shows the series of reflections $h0l$ and $h01$, but these should be extincted in $P2_1/c$ ($h0l: l = 2n$).

We have carefully evaluated our data as well as performed additional experiments to assess the symmetry. While it is true that the data does contain multiple scattering, as it is very hard to avoid for electron diffraction data as the reviewer already stated, all our results suggest that the $P2_1/c$ symmetry is the proper choice. By acquiring electron diffraction patterns slightly tilted away from zone axis the multiple scattering will be reduced and the systematic extinctions will show up clearly. In Figure 1 below, electron diffraction patterns have been acquired along the $[100]$ direction, as well as tilted $\sim 8^\circ$ away from the main zone axis around the b^ and c^* -axes. While the pattern acquired along the $[100]$ direction is heavily affected by multiple scattering (where no extinctions are visible), after tilting the crystal $\sim 8^\circ$ from the zone axis, clear extinction shows up both along the b^* and c^* -axes. Among the primitive monoclinic space groups there is just one extinction symbol which is consistent with extinctions along two perpendicular axes leaving us with just one unique choice of symmetry for DCPM, namely $P2_1/c$ (No. 14). The validity of the space group is now also supported by quantum mechanical calculations that find no imaginary phonon modes for the structure optimized within this symmetry.*

Since the morphology of the DCPM crystals is plate-like, the crystals will likely pass close to the $0kl$ plane of the reciprocal lattice and hence the cRED diffraction data will be influenced by dynamic effects.

Evaluation of several cRED data sets shows that reflections belonging to the $h0l$ and $h01$ families appear with different intensities. For data sets that do not pass close to the main zone axes, e.g. the data set shown in Figure 2 below, the given reflections appear absent or very weak. This is clearly consistent with a c -glide in the structure of DCPM. This shows that multiple scattering is responsible for the observed intensities appearing in Figure 1d of the original manuscript showing the $h0l$ family of reflections.

The multiple scattering does contribute to the high R -value, as is common for refinements of 3D electron diffraction data. It should, however, not raise concerns related to the correctness of our structure model as the structure is also consistent with the XRPD data and theoretical calculations.

Figure 1 and 2 below have been added to the Supporting information of the manuscript (as Figure S4 and S5) in order to confirm the symmetry of DCPM to be $P2_1/c$, as have details of the quantum mechanical calculations used to further confirm the validity of the space group.

Further details on the symmetry deduction, structure solution and refinements have been added to the supporting information.

Figure 1. Three selected area electron diffraction patterns from a crystal of DCPM. The middle pattern is acquired along the $[100]$ direction and hence affected by dynamic scattering. The pattern to the left is tilted away from the $[100]$ direction by $\sim 8^\circ$ around the b^* -axis. It is clear that systematic extinctions appear along the b^* -axis as reflections with odd k -indices are now absent. Following the parallel procedure, the pattern to the right reveals clear extinctions for reflections in the $00l$ family with odd l -indices when the crystal is tilted around the c^* -axis by $\sim 8^\circ$. This is clearly consistent with the symmetry $P2_1/c$.

Figure 2. A section through the reconstructed reciprocal lattice of a cRED data set from DCPM including the $h0l$ family of reflections. In this data set reflections with odd l -indices are very close to extinct. This is consistent with a c -glide in the crystal structure of DCPM. Horizontal lines with odd l -indices are marked by red arrows. The

reason that this data set shows clearer extinctions is that the crystal did not pass through any main zone axis during the data collection. Note that traces of diffraction from small ice crystals formed during the data collection at liquid nitrogen temperature can be observed (colored in light blue).

Additional questions:

- How many parameters have been refined using 563 independent reflections?

The least squares refinement of the DCPM structure against the cRED data was performed with isotropic atomic displacement parameters using a total of 29 parameters. Given the number of independent reflections being 563 this is well within reasonable limits.

Further details regarding the refinements against cRED data have been added to the supporting information and Table S1.

- Uncertainties for a, b, c, beta are missing in Table S1.

The errors for the unit cell parameters from cRED data can be estimated in different ways. During indexing and data extraction using XDS the unit cell was determined to be: $a=8.139(14)\text{\AA}$, $b=6.826(3)\text{\AA}$, $c=8.221(8)\text{\AA}$, $\alpha=90^\circ$, $\beta=91.42(23)^\circ$, $\gamma=90^\circ$ including estimated standard deviations. However, these uncertainties do not take systematic errors into account and the real error will be larger. Based on 8 cRED data sets collected from DCPM the average unit cell was determined to be $a=8.024\text{\AA}$, $b=6.989\text{\AA}$, $c=8.234\text{\AA}$, $\alpha=90^\circ$, $\beta=92.19^\circ$, $\gamma=90^\circ$ with estimated standard deviations of 0.13\AA , 0.13\AA , 0.08\AA , 0° , 0.82° and 0° , respectively. This is a better representation of the errors of the unit cell parameters in the cRED data. Due to the significant errors in unit cell determination, the final refinements of the structure of DCPM against the cRED data was performed based on the unit cell as determined from Rietveld refinement.

Table S1 was updated accordingly.

Unfortunately, the treatment of the x-ray powder data has as well some flaws.

- The presented X-ray powder diffraction data is typical for a layered structure. Rietveld refinement provided in the SI confirms the structure but the achieved fit is not explaining all intensities leading to an somewhat acceptable residual of 8.6 %. There are smaller peaks in Fig. 5 at e.g. 16.0° ; 18.2° ; 23.7° etc, so an impurity that is not being discussed. The areas like 28 to 31° do not fit very well. It would be important to use the stacking error model included in Topas to improve FitQuality. Stacking faults as indicated by ED data should definitely be taken into account.

Which reflections from cRED data (match with PXRD?)

In order to explore the evolution of the DCPM phase during crystallization we have now performed a time-resolved study of its formation with powder X-ray diffraction data acquired after 3h, 5h, as well as 7h. The main features of the XRPD patterns are the same in all three patterns showing that the DCPM phase is dominating the time sequence (see Figure 3 below). However, some of the fine details of the patterns do change during the time series. A comparison between the three samples of the time series, as well as the XRPD data of DCPM presented in the original submission, are compared in Figure 4 below. The peaks at 16.0° ; 18.2° ; 23.7° in the original data, as the reviewer pointed out, are no longer present in the newly synthesized batch of DCPM; see Figure 4 a and b. Some additional discrepancies were present in the domain $29^\circ - 34^\circ$ of original refinement. The peaks at $\sim 29^\circ$, as well as 30.3° , are close to diminished in the 3h and 5h patterns of Figure 3c below. In the tail of the 121 reflection at 30.9° some diffuse scattering intensities are present in the range $31^\circ - 32^\circ$.

The diffuse scattering is surrounding the 20-2 reflection at 31.6° . This scattering changes character through the three samples of the time series but now centers its intensity around 31.6° which is the position of the 20-2 reflection. At 9.97° a weak reflection appears in the newly synthesized batch which was not present in the pattern presented earlier. This is attributed to an unknown impurity phase. Since it is just present in this synthetic batch, we conclude that it is not a characteristic peak of the DCPM phase.

Figure 3. XRPD patterns acquired after 3h, 5h as well as 7h of crystallization of DCPM.

Figure 4. Comparison between the new XRPD patterns acquired after 3h, 5h and 7h of crystallization as well as the XRPD data of DCPM presented in the earlier submission (labeled “old”). All four patterns show great similarity, though some differences can be observed in the four angular ranges presented in panels (a)-(d).

A new Rietveld refinement with a significantly improved fit was performed based on the data after 3h in the time series which contains less effects due to the diffuse scattering in the 30° - 32° region. The Rietveld refinement was performed without any restraints on bond distances or the geometry of the structure. A fourth-order spherical harmonic correction for the preferred orientation of the crystals was introduced thanks to the reviewer's comments, resulting in a significantly improved fit as can be seen in Figure 5. The refinement was performed with 7 parameters to fit the background and a pseudo-Voigt peak shape function to fit the peaks.

Although the diffuse intensities between 30° - 32° have been reduced significantly there are still some remaining issues to fit their intensities with the pseudo-Voigt peak shape function. The diffuse scattering is surrounding the 121 and 20-2 reflections in both the cRED data as well as the XRPD data, see Figure 6 below. The diffuse scattering is most likely occurring due to imperfections in the stacking of the layers. The weak interaction between the layers however allows for numerous theoretically possible transitions between layers. In order to model the diffuse scattering either using DIFFaX simulations or performing refinements including stacking faults in Topas a transition model is needed. Given that the diffuse scattering is minor, and that there are few structural restrictions on the layer connectivity, it would not be justified to present a definite model for the disorder other than that the material does contain some stacking disorder introducing imperfections in the connectivity of the layers.

Figure 5. Rietveld refinement of DCPM phase against the new XRPD data. The fit converges well with a residual R_{wp} of 6.667%. The range 17° - 70° is enlarged in the inset for clarity. In the domain 29° – 34° there are some issues with the peak shape of some reflections. This can be attributed the presence of some diffuse scattering also observed in the cRED data.

Figure 6. Part of the reconstructed three-dimensional reciprocal lattice from data collected from DCPM shows some diffusely scattered streaks running along the a^* -axis. The elliptical feature arises due to scattering from small ice crystals formed when the sample was cooled during data collection. The 121 reflection (in blue) shows diffuse intensities extending along the a^* -axis to both higher and lower scattering angles. The $h02$ lines also exhibits diffuse intensities along the a^* -axis. (b and c) Sections through a full voxel-by-voxel reconstruction reveals weak diffuse scattering around the 121 reflection in the $hk1$ section (a), some weak diffuse scattering is also present along the $h02$ line in (c). (d) Enlarged section of the XRPD pattern of DCPM between 27.4° and 35.7° of 2θ . In this regime, some diffuse intensities are present which cannot be fitted well with the pseudo-Voigt peak shape. The corresponding d -spacings as well as indices of the reflections are consistent with the diffuse scattering present in the cRED data.

The geometry of the experiment (capillary or flat sample) is not mentioned?

The X-ray powder diffraction (XRPD) patterns were recorded using an X-ray diffractometer (Bruker D8 ADVANCE) in reflectance Bragg-Brentano geometry employing Ni filtered Cu K α line focused radiation ($\lambda=1.54178 \text{ \AA}$) at 3000W (40 kV, 40 mA) power. Each sample was measured with a 2θ rate of $10^\circ/\text{min}$ for normal characterization, and $1^\circ/\text{min}$ for refinement.

The plate-like morphology in combination with the fact that the data was collected in reflection mode imposes preferred orientation effects on the intensities of the XRPD pattern. These effects were corrected for by a spherical harmonic correction introducing 9 parameters to the refinement.

The details regarding the collection of XRPD data have been added to the Methods section.

- Uncertainties for a, b, c, beta in the Rietveld Fit are missing in Table S2.

Uncertainties for the unit cell parameters have now been added to Table S2.

- In supporting info the wavelength 1.54178 \AA is stated while in Table S2 it is 1.5406 \AA ?

We thank the reviewer for pointing this out. We have now corrected the wavelength in Table S2 to 1.54178 \AA .

Comparing both methods:

- The atomic positions derived by ED refinement and Rietveld refinement are both missing. How similar are the resulting structures? Are there significant differences in the coordination polyhedra? What are the uncertainties of the atomic positions of the APDs in the Rietveld / ED refinement?

The cif-file describing the refined structure was unfortunately not submitted along with the original manuscript due to a mistake, as was the reference to the structure deposited in the CCDC data base. We apologize for this mistake.

The structures after cRED and Rietveld refinement are very similar. The average difference in atomic position between the two structures is 0.155 \AA , with the largest difference being 0.253 \AA for one of the oxygen atoms of the phosphate ion, see Figure 7. The Ca^{2+} ion coordinates to six oxygen atoms of phosphate ions in both the structure refined against cRED as well as Rietveld data; see Figure 8. The average difference for the Ca-O distances between the two structures is 0.04 \AA with a largest difference of 0.08 \AA . The distance between the Ca^{2+} ion and the water molecule is 2.42 \AA in the cRED data and 2.30 \AA for the Rietveld refinement. The average difference for the P-O distances of the phosphate ion is 0.11 \AA ; see Figure 8 for further details.

Uncertainties for atomic coordinates, as well as atomic displacement parameters, can be found in the attached cif-files.

In order to further evaluate the feasibility of the structure of DCPM first principles quantum mechanical calculations were performed. Hydrogen atoms were inserted into the structure for a variety of different initial positions for both the HPO_4^{2-} ions and water, followed by geometry optimization and annealing via molecular dynamics. A consistent picture emerges in which two chains of hydrogen bonded HPO_4^{2-} anions run parallel to the b axis, with the OH groups pointing in opposite directions in each chain, see Figure 7. Attempts to transfer the proton along the chain (i.e. from OH...O to

O...HO) led to spontaneous return of the hydrogen to its initial site, indicating a strong preference for the original order. Water coordinates to one calcium via oxygen and is simultaneously able to hydrogen bond to two different anions. All hydrogen positions are commensurate with the space group without the need for partial occupancy or disorder.

Figure 7. Comparison of the structures of DCPM after refinement against cRED data (left), XRPD data (middle) as well as the DFT optimized structure (right). The structures are viewed along the [001] direction. Green represents Ca, purple P, red O and white H.

Figure 8. Comparison of the coordination around the calcium ion and phosphorous atom for the structures after refinement against cRED (left) and XRPD (right) data. Green represents Ca, purple P and red O. The oxygen atom to the top right of the Ca atom is the water molecule in the vicinity of the Ca ion. Bond lengths and distances indicated are in Ångströms.

Figures 7 and 8 have been added to the supporting information in order to show the similarity between the model refined using cRED and XRPD data.

- Why is b smaller in the Rietveld refinement than in the ED, while for a and c it is larger (Table S1 and S2)? If it were a pure camera length effect I would expect a consistent trend.

The unit cell parameters as determined from electron diffraction data contain significant uncertainties as was discussed in the answer to an earlier comment. In order to obtain a fair measure of the errors an average unit cell based on 8 data sets was calculated. Comparing the average unit cell to the one determined from Rietveld refinement all three unit cell axes are now longer for electron diffraction data. For the a-axis the difference is however just 0.2% whereas for b and c it is 2.7% and 5.3%, respectively, and for β it is 0.65°. The main contribution to the differences is most likely the measurement error although effects of the vacuum and electron irradiation cannot be excluded.

Because of the known uncertainties of the unit cell parameters as determined from electron diffraction data all refinements were performed using the unit cell as determined from XRPD data.

A corresponding note has been added to the main text.

Reviewer #2 (Remarks to the Author):

The authors carried out a huge data analysis. It follows from the conclusion made in the peer-reviewed work that the authors determine the existence of a phase precursor during the mineralization of dicalcium phosphate dihydrate (DCPD) from amorphous calcium phosphate (ACP). In this work, a new form of calcium phosphate (CaP) is introduced, named Konhaitite (after the two places of its discovery, Konstanz and Shanghai), which is dicalcium phosphate monohydrate (DCPM, $\text{CaHPO}_4 \cdot \text{H}_2\text{O}$). This, of course, is important and necessary knowledge for a deeper understanding of the process of biomineralization. Such research deepens our understanding of the processes and mechanisms of calcium phosphates formation and is consistent with current scientific trends. The work was carried out at a high methodological and scientific level, using a complex of modern methods and instruments.

We thank the reviewer for the positive comments.

The disadvantages of the article include the following points.

The abstract does not give a complete picture of the study and the content of the article. The authors have studied a large amount of material, but historical references should be included in the introduction. It is more important in the annotation to describe the features of the method for obtaining DCPM material and its main differences from previously applied in the annotation, it is important to show what methods were used to study the resulting structure and its distinctive characteristics. Twice talking about the importance of material at the beginning and end of an abstract is also unnecessary.

We thank the reviewer for pointing this out.

The abstract has been revised in the new version, so as to describe the synthesis and include the methods for the structure determination. Also, we avoid pointing out the importance of our finding twice, and changed the last sentence of the abstract accordingly.

In the introduction, historical references (refs. 2-7 in the revised version) to the mentioned CaPs have now been included.

The significance of the study conducted by the authors is undeniable, but in the main part, the methods for obtaining and fixing the structure in question are not sufficiently discussed. The investigated phase is a transitional, difficult to fix, therefore, the features of obtaining and research would deepen our

understanding of the results. The authors did not specify either the conditions for the creation of this phase, except that it was obtained by the methanol / water approach, nor the fixation conditions to prevent further transitions and the possibility of research. The reasons for choosing this method using a mixture of methanol and water are not substantiated or described

In the revised manuscript, the method of DCPM preparation is now discussed in more detail.

As mentioned above, the preparation protocol relies on (1) amorphous calcium hydrogen phosphate (ACHP) as the precursor (B. Q. Lu et al. Crystal Growth & Design 19, 3030-3038, 2019), and (2) water-poor environments for the transformation. DCPM was previously neither observed during the dehydration of DCPD nor the hydration of DCP in H₂O. We also failed to obtain DCPM from other amorphous calcium phosphates (with Ca/P atomic ratios higher than 1.0). This underpins the importance of ACHP as the precursor for DCPM formation. Besides, H₂O is required for the crystallization of DCPM from ACHP, as the amorphous precursor did indeed not crystallize in dry air (humidity < 5 %) for at least one month, or transformed into other phases in more water-poor methanol. Third, the time for controlling the ACHP transformation should be appropriate.

Although prepared by ACHP transformation in a mixture of methanol (32.5 mL) and water (5 mL), DCPM is actually an intermediate phase between ACHP and DCP in this case. As indicated by XPRD and IR analyses, DCPM can be formed as early as 2 min, and kept for 7 h. But after 24 h, while a small fraction of DCPM can still be detected, the majority phase became DCP. In humid air, DCPM was the majority phase of the transformation of ACHP after 4h, and ultimately transformed into DCP after 24 h. In water (Figure 5 in the main text), DCPM transformed into HAP.

The reason for choosing ACHP (B. Q. Lu et al. Crystal Growth & Design 19, 3030-3038, 2019) as the precursor was mainly due to their similar chemical compositions (CaHPO₄·H₂O). Water-poor environments appear to be necessary to (1) induce the ACHP transformation, whereas (2) too much water will lead to other phases such as DCPD (B. Q. Lu et al. Crystal Growth & Design 19, 3030-3038, 2019). At the specific conditions detailed in our paper, DCPM can emerge as the intermediate phase between ACHP and DCP/DCPD, or HAP.

The transformation of DCPM can be prevented by storing it in dry (humidity < 5%) or cold (<-20°C) environments, or using organic stabilizers such as sodium citrate, as discussed already in the first version of the manuscript.

Corresponding details have been added to the main text and the Methods section.

the authors also noted that this phase of the DCPM is formed under conditions of high humidity during the transition of ACP to DCPD, but with less crystallinity. It is difficult to draw such conclusions on the basis of Fig. 3, since a very small scale was chosen and the magnifications were not made for all options (only for 12 h.).

We believe there may be a misunderstanding regarding on how we conclude that there are differences in the crystallinity. Instead of Fig. 3 (Fig. 4 in the revised version), we assessed this based on the XPRD patterns (Fig. S12 in the revised version) of the two samples prepared by two different methods. The somewhat broader reflections of DCPM formed in humid air indicate lower crystallinity.

We checked the text and added the missing cross-reference to Fig S12, so as to avoid misunderstanding.

The unconditional merit of the authors lies in the fact that they use a large number of existing methods and modern instrumentation. However, perhaps, the authors chose an incorrect presentation of the results. The graphs presented in the work are very difficult to compare with the conclusions made by the authors. For logical conclusions based on the data provided, it would be convenient to show the basic structural and chemical characteristics of the newly formed phase in the form of a table in comparison with the previously known DCPD, ACP, and octacalcium phosphate.

We thank the reviewer for the suggestion.

In the revised manuscript, a comparison of DCPM with other CaPs has been added (tables S3 and S5).

Editorial Note: Reviewer #1 was unable to assess the revision so a new reviewer was asked to assess the responses to their comments, this reviewer is numbered reviewer #3

Reviewers' Comments:

Reviewer #2:

Remarks to the Author:

The comments that I received were sufficient for the publication in Nature Communications.

Vladimir S. Komlev

Reviewer #3:

Remarks to the Author:

This review report is strictly in response to the requirements of the editor.

Reviewer 1 raised two key issues and some minor questions/suggestions. The authors have properly addressed the minor questions, and followed the suggestions. I will comment on both two key issues in detail.

1. The significance of the fabrication, stabilization and transformation of the new CaP phase. While it is true that DCPM is just a new form of dicalcium phosphate, the possibility of DCPM as a potential implant and drug delivery material has been clearly demonstrated by the cell viability tests. It seems DCPM is thermodynamically more unstable than DCP and DCPD, indicating it is also more active, so I am not surprised that DCPM with a medium amount of water has the best capacity of absorbing dye and drug molecules. This reminds me of amorphous calcium carbonate, which is also unstable and has proved a good drug delivery material. I would also expect a different cell behavior on a DCPM surface, and a different degradation kinetics of DCPM coating on bone implants. There are more to investigate about this new CaP phase, which is not possible to cover in one single paper. In this manuscript, the authors have already provided enough evidence. A remaining concern is the low stability of DCPM in aqueous environment. The authors used sodium citrate to stabilize DCPM but they observed for only two hours, which were far from enough. Polyelectrolytes may also help.

2. The extinction condition of P21/c.

Reviewer 1 questioned the presence of h03 and h01 series in Fig. 1d, because they should both extinct for P21/c space group (when l is odd, h0l should be very weak or disappear). I am not sure if Fig. 1d remains the same as the previous version (by the way, the authors neither noted Fig. 1d in the figure caption, nor discussed 1d in the main text as they had done in the response letter). Yet the question was quite clear.

The authors successfully followed a simple method to exclude the spots derived from multiple scattering by slightly varying the orientation. In Figure 2 in the response letter, the odd-l h0l series vanish, which means the odd-l h0l spots that should not be very strong in previous Fig. 4d were actually multiple scattering spots, and the diffraction pattern well satisfies the P21/c space group. Along with the fact that the authors had revised the data and fitting according to the comments, the structures has been clearly described.

Point-by-point replies

Our replies appear indented and italicized, corresponding changes in the manuscript and the SI are highlighted in yellow.

Reviewer #2 (Remarks to the Author):

The comments that I received were sufficient for the publication in Nature Communications.
Vladimir S. Komlev

We thank Prof. Komlev for his positive assessment.

Reviewer #3 (Remarks to the Author):

This review report is strictly in response to the requirements of the editor.

Reviewer 1 raised two key issues and some minor questions/suggestions. The authors have properly addressed the minor questions, and followed the suggestions. I will comment on both two key issues in detail.

1. The significance of the fabrication, stabilization and transformation of the new CaP phase.

While it is true that DCPM is just a new form of dicalcium phosphate, the possibility of DCPM as a potential implant and drug delivery material has been clearly demonstrated by the cell viability tests. It seems DCPM is thermodynamically more unstable than DCP and DCPD, indicating it is also more active, so I am not surprised that DCPM with a medium amount of water has the best capacity of absorbing dye and drug molecules. This reminds me of amorphous calcium carbonate, which is also unstable and has proved a good drug delivery material. I would also expect a different cell behavior on a DCPM surface, and a different degradation kinetics of DCPM coating on bone implants. There are more to investigate about this new CaP phase, which is not possible to cover in one single paper. In this manuscript, the authors have already provided enough evidence.

We thank the Reviewer for the positive assessment.

A remaining concern is the low stability of DCPM in aqueous environment. The authors used sodium citrate to stabilize DCPM but they observed for only two hours, which were far from enough. Polyelectrolytes may also help.

We thank the Reviewer for the constructive remark, and following the suggestion, we further studied the stabilization of DCPM in a solution of sodium polyacrylate (pH 7.0), a polyelectrolyte often used as a mineralization additive. Indeed, as shown in Figure 5a, sodium polyacrylate stabilizes DCPM for at least 13 hours (the XPRD pattern does not differ from that of the initial DCPM), much longer than sodium citrate (2 h). A corresponding, brief discussion has been added on page 12 of the revised manuscript, the methods section has been updated (pages 14, 15).

We agree that the stabilization of DCPM is a very important and interesting subject, which should be studied in detail in the future.

2. The extinction condition of P21/c.

Reviewer 1 questioned the presence of h03 and h01 series in Fig. 1d, because they should both extinct for P21/c space group (when l is odd, h0l should be very weak or disappear). I am not sure if Fig. 1d remains the same as the previous version (by the way, the authors neither noted Fig. 1d in the figure caption, nor discussed 1d in the main text as they had done in the response letter). Yet the question was quite clear.

The authors successfully followed a simple method to exclude the spots derived from multiple scattering by slightly varying the orientation. In Figure 2 in the response letter, the odd-l h0l series vanish, which means the odd-l h0l spots that should not be very strong in previous Fig. 4d were

actually multiple scattering spots, and the diffraction pattern well satisfies the $P2_1/c$ space group. Along with the fact that the authors had revised the data and fitting according to the comments, the structures has been clearly described.

We thank the Reviewer for the positive assessment.

We apologize that the label of Figure 1d (the figure was updated in the previous revision) in the corresponding caption was lost, which has been corrected in the revised version (page 4).

The detailed discussion of Figure 1d (also referring to Figure S6), was presented in the caption below Figure S6, and cross-referenced on page 6 of the main text. On the same page, the figure reference was mistakenly 2d instead of 1d, however, which has been corrected in the revised version.

The detailed discussion of the determination of the $P2_1/c$ space group was presented in the captions of Figures S3 and S4, which were cross-referenced in the caption of Figure 1. So as to avoid misunderstandings and further strengthen the conclusion, we have furthermore added a brief Comment Section to the SI (page 18), which reflects the related discussion in our previous response letter. In the main text of the revised manuscript, Figure 1, S3, S4 and the Comment Section in the SI are now cross-referenced on page 6.

Other changes and corrections:

The unit cell parameters of the known dicalcium phosphates (i.e., dicalcium phosphate dihydrate, DCPD, and dicalcium phosphate, DCP) have been updated according to the latest references (main text, page 8; SI, table S3). In the previous version, we erroneously referred to a choice of unit cell parameters for the same structures, which are not commonly used.

Reviewers' Comments:

Reviewer #3:

Remarks to the Author:
Comments

The authors had well answered my comments and made suitable revisions. I think this revised manuscript can be accepted as is.

Point-by-point replies

Our replies appear indented and italicized, tracked changes are preserved in the manuscript.

REVIEWERS' COMMENTS:

Reviewer #3:

Comments

The authors had well answered my comments and made suitable revisions. I think this revised manuscript can be accepted as is.

We thank the reviewer for the positive assessment.